# Stability of Biomimetically Functionalised Alginate Microspheres as 3D Support in Cell Cultures

**DOI:** 10.3390/polym14204282

**Published:** 2022-10-12

**Authors:** María Inmaculada García-Briega, Joaquín Ródenas-Rochina, Luis Amaro Martins, Senentxu Lanceros-Méndez, Gloria Gallego Ferrer, Amparo Sempere, José Luís Gómez Ribelles

**Affiliations:** 1Centre for Biomaterials and Tissue Engineering (CBIT) Universitat Politècnica de València, 46022 Valencia, Spain; 2Biomedical Research Networking Centre on Bioengineering, Biomaterials and Nanomedicine (CIBER-BBN), 46022 Valencia, Spain; 3Centre of Physics, Universidade Do Minho, 4710-057 Braga, Portugal; 4BCMaterials, Basque Centre for Materials, Applications and Nanostructures, UPV/EHU Science Park, 48940 Leioa, Spain; 5IKERBASQUE, Basque Foundation for Science, 48013 Bilbao, Spain; 6Grupo de Investigación en Hematología, Instituto de Investigación Sanitaria La Fe (IISLAFE), 46026 Valencia, Spain; 7Centro de Investigación Biomédica en Red de Cáncer (CIBERONC), Instituto Carlos III, 28029 Madrid, Spain

**Keywords:** alginate, microspheres, stability, layer by layer, multiple myeloma

## Abstract

Alginate hydrogels can be used to develop a three-dimensional environment in which various cell types can be grown. Cross-linking the alginate chains using reversible ionic bonds opens up great possibilities for the encapsulation and subsequent release of cells or drugs. However, alginate also has a drawback in that its structure is not very stable in a culture medium with cellular activity. This work explored the stability of alginate microspheres functionalised by grafting specific biomolecules onto their surface to form microgels in which biomimetic microspheres surrounded the cells in the culture, reproducing the natural microenvironment. A study was made of the stability of the microgel in different typical culture media and the formation of polyelectrolyte multilayers containing polylysine and heparin. Multiple myeloma cell proliferation in the culture was tested in a bioreactor under gentle agitation.

## 1. Introduction

Alginate (ALG) is a polysaccharide composed of linear block copolymer chains with β-D-mannuronic acid (M) and α–L-guluronic acid (G) blocks obtained from brown algae. Alginate hydrogels are commonly prepared by ionic cross-linking using divalent cations, particularly calcium (Ca^2+^) ions. Monomeric units of G blocks cooperatively bind to a calcium cation, and these bonds act as crossover points between the chains forming a reversible hydrogel. ALG hydrogels are highly biocompatible and have a low interaction with cells as they lack adhesion sequences [1,2], which is the reason why alginate hydrogels have been proposed for in vitro and in vivo applications, especially for drug delivery [3], 3D cell culture [4,5] and bioprinting [2,6]. On the other hand, industrial processes for the extraction of alginate from brown algae or its biotechnological production using certain bacteria make high-purity alginates available for medical applications [7].

The fact that alginate ionic cross-linking is a reversible process generates great interest in cell culture in a 3D environment [4]. For example, chondrocytes can be cultured encapsulated in the alginate gel and in a chondrogenic medium and extracted from the hydrogel using a calcium chelator [8,9]. The cells thus obtained can be transplanted into the organism in a scaffold or hydrogel with controlled biodegradation. Alginate can be combined with ceramic particles to form a macroporous scaffold [10]. A 3D environment has recently been proposed for cell culture through an agglomerate of microspheres suspended in an aqueous medium identified as a microgel [11,12,13,14]. The cells are seeded into the space between the microspheres, whose surface can be functionalised to provide cell adhesion sequences [14,15,16], while the microspheres themselves can support the release of growth factors during culture [17,18]. The same strategy can be used for in vivo tissue regeneration [19].

With regard to the production of microspheres to form microgels, alginate has the advantage over other hydrogels that the cross-linking kinetics is fast and does not require organic solvents or other synthetic reagents. Moreover, the surface of the alginate microspheres is negatively charged, which will facilitate the deposition of biomolecules of interest by forming multilayers of polyelectrolytes layer by layer, giving the microgel a biomimetic character. Nevertheless, for tissue engineering applications, alginate hydrogels have the drawback of slow, incomplete and uncontrollable biodegradation [1,20]. The cross-linking density of alginate hydrogels produced by ionic cross-linking is continuously reduced by the loss of calcium ions in an aqueous medium, even at pH 7, thus reducing the mechanical properties of the gel, which in any case are quite limited [21]. The use of alginate microspheres to form a microgel for cell culture raises the question of its long-term stability when the gel is immersed in a culture medium. The thermodynamic equilibrium determines the swelling capacity of the alginate hydrogel in an aqueous medium and the migration of calcium ions from the gel to the medium. The alginate can be cross-linked by covalent bonds to increase the stability of the material during cultivation, which requires modifying the chemical structure of the polysaccharide chain to introduce functional groups [22]. Another way of stabilising alginate microspheres is to coat them with multilayers of polyelectrolytes in a layer-by-layer (LbL) procedure [23]. The coating that forms on the microspheres’ surface has a wide variety of uses, ranging from simple functionalisation to use as a molecular filter, besides stabilising the gel itself. Alginate microspheres with an LbL alginate-polylysine coating were proposed to encapsulate pancreatic islets in the so-called “bioartificial pancreas” [24]. Alginate microspheres have also been coated with alginate/chitosan multilayers. The membranes formed with polyelectrolyte multilayers can liquefy the alginate core by using a calcium chelator to form microcapsules. These have been proposed as microreactors, since by encapsulating the cells in the alginate suspension that forms the microsphere and eliminating the alginate cross-linking, only cells and the specifically designed microenvironment remain inside the membrane [25]. Membranes have also been proposed for protein delivery [26].

In this work, alginate microspheres were functionalised by coating their surface with biomolecules from the extracellular matrix of the tissue of interest for use as a 3D support in cell culture. It is particularly intended to apply them for application to the development of a disease model for multiple myeloma. The study focused on the stability of the microspheres in the different liquid media in which they will be immersed, either in a cell culture or in an LbL processes. The microgel’s stability under natural culture conditions and the biological response were also analysed.

## 2. Hypotheses

The equilibrium swelling of the functionalised alginate microspheres immersed in an aqueous liquid medium depends on thermodynamic equilibrium criteria and therefore on the salts dissolved in the liquid. In many cell culture mediums, the dynamic process in which the gel loses cross-linking points reduces its mechanical consistency, accompanied by increased water absorption capacity and the delivery of alginate chains to the liquid medium, a process that ultimately leads to the dissolution of the microsphere. The multilayer polyelectrolyte coating can maintain the microsphere’s integrity even when the alginate core becomes liquid. However, the microsphere may start to swell, a process that compromises its integrity, depending on the aqueous medium and the elastic properties of the coating.

## 3. Materials and Methods

### 3.1. Materials

A solution of sodium alginate from brown algae (Sigma-Aldrich, St. Louis, MO, USA) and calcium carbonate (Mw 100.09 Da, Scharlab, Barcelona, Spain) in ultra-pure water was used as a discontinuous medium. The continuous medium was composed of extra-virgin olive oil and soy lecithin. Acetic acid (Sigma-Aldrich) was also used for reticulating the microsphere. Different coatings were used to functionalise the microspheres. Chitosan (Mw 190,000–375,000 Da and 75% deacetylation), heparin sodium salt from porcine intestinal mucosa (203 ud/mg), sodium chloride (Mw 58.44 Da) and poly-L-Lysine hydrobromide (Mw 47000 Da), in addition to alginate, were purchased from Sigma-Aldrich. N-hydroxysuccinimide (Mw 115.09 Da) and glycine (Mw 75.07 Da) (Sigma-Aldrich) were used to cross-link the coatings, as well as glutaraldehyde (concentration of 25% and Mw 100.12 Da, Scharlab) and N-(3-dimethylaminopropyl)-N′-ethylcarbodiimide (EDC, Mw 191.70 Da, Iris Biotech GmbH, Marktredwitz, Germany).

A commercial B1000 Blyscan kit from Bicolor was purchased to characterise the microspheres. The stability assay was performed with ethylenediaminetetraacetic acid 0.5 M (EDTA, from Invitrogen, Waltham, MA, USA), and the culture media RPMI 1640 (Sigma-Aldrich), Dulbecco’s Modified Eagle’s Medium High Glucose with Phenol Red w/o L-Glutamine nor Sodium Pyruvate (DMEM R, Biowest, Nuaillé, France) and Dulbecco’s Modified Eagle’s Medium Low Glucose without Phenol Red w/o L-Glutamine w Sodium Pyruvate (DMEM W, Gibco, Billings, MT, USA) were used. Moreover, Dulbecco’s Phosphate-Buffered Saline with calcium and magnesium (PBS +/+, Biowest) and without (PBS −/−, Gibco) were used.

### 3.2. Production of Alginate Microspheres

Alginate microspheres (moALG) were produced using a polydimethylsiloxane, Sylgadr 184 flow-focusing microfluidic device with a channel size of 500 × 500 µm. The discontinuous phase was composed of sodium alginate dissolved at 1% *w*/*v* in Milli-Q water containing calcium carbonate dispersed at 0.25 M. The continuous phase was based on olive oil with 1% *w*/*w* soy lecithin as a surfactant. Microspheres are formed by the immiscibility of water in oil and will cross-link when they encounter what we call a “collector medium”, which is obtained by adding 2% *v*/*w* acetic acid to the continuous phase. Alginate cross-linking is ionic since calcium carbonate dissociates in an acid medium, releasing calcium (Ca^2+^) ions that cross-link the alginate microsphere from the inside. A diagram of the production system is shown in Figure 1.

### 3.3. Layer by Layer Functionalisation

Two surface functionalisations of alginate microspheres were conducted. Both series were produced by the layer-by-layer (LbL) technique, in which oppositely charged polyelectrolytes are deposited alternately. The surface coating is obtained by immersing the microspheres in the two polyelectrolytes, with rinses in between, as shown in the diagram in Figure 2. The method proposed by Li et al. was adapted to obtain microspheres coated with alternate layers of chitosan (CHI) and heparin (HEP), known as moCHI/HEP [27]. Briefly, moALG microspheres were dispersed in a 0.1% *w*/*v* CHI solution with 0.25 M NaCl and 0.1 M acetic acid, at pH 5.5, under gentle stirring for 15 min. They were then washed twice in water at pH 5.5 and 0.25 M NaCl for 5 min each, and the microspheres were transferred to a 0.1% *w*/*v* HEP solution with 0.25 M NaCl at pH 5.5 for 15 min, after which the process was repeated until three bilayers were formed, ending in a layer of HEP. The last step was immersion in a 0.06 M EDC, 0.03 M NHS and 0.25 M NaCl solution at pH 7 and stirred overnight to cross-link the coating, using an adaptation of the protocol described by Li et al. [28].

The second Polylysine (PLL) and ALG functionalisation process (moPLL/ALG) was adapted from that described by Correia et al. [29]. Briefly, moALG microspheres were dispersed in a PLL 0.05% *w*/*v* solution at pH 7 with 0.15 M NaCl for 10 min. Then the microspheres were washed twice with pH 7 water and 0.15 M NaCl for 5 min each and immersed in an ALG 0.05% *w*/*v* solution at pH 7 with 0.15 M NaCl for 10 min, after which the microspheres were twice washed with water at pH 7 and 0.15 M NaCl for 5 min each. The process was repeated twice to form two bilayers, ending with ALG. No cross-linking took place in this case due to the strong electrostatic interaction between PLL and ALG in the LbL coating [30].

### 3.4. Microsphere Characterisation

Optical microscopy, ImageJ imaging software and Excel and Graphpad statistical software were used to evaluate the shape and size of the microspheres. The internal alginate structure in the microspheres was visualised by cryoFESEM, and images were taken from uncoated samples at an accelerating voltage of 1 kV. Fourier transform infrared spectroscopy (FTIR) and thermogravimetric analysis (TGA) were used to determine their composition. The TGA parameters for the measurements were a heating rate of 10 °C/min and a nitrogen flow rate of 50 mL/min. The samples had an initial weight of approximately 55 mg, and the analysis was carried out at temperatures above 200 °C, when no water remained in the microsphere. Blysscan (B1000 Blyscan kit from Bicolor, Beverly Hills, CA, USA) and ninhydrin colorimetric assays were used to quantify the amount of HEP and PLL in the coating, the latter based on a protocol described by Algieri et al. [31].

### 3.5. Equilibrium Swelling

Equilibrium swelling is a quantitative method of determining the liquid media in which the synthesised microspheres maintain their cross-linked structure and remain insoluble. Their stability was tested in different culture media (RPMI 1640, DMEM R, DMEM W, PBS+/+, and PBS−/−) and in water at different pH to assess the influence of pH on cross-linking stability. This test also verifies that the coating allows the interior to liquefy by immersing all types of microspheres in EDTA, which is a calcium chelator. The test assesses variations in diameter in each condition by means of the Feret diameter, measured by ImageJ on optical microscopy images and GraphPad statistics.

### 3.6. In Vitro Assay

An in vitro assay was performed on days 3 and 5 to test cell proliferation with and without microspheres. The culture platform comprised 60,000 cells of the RPMI 8226 cell line in 200 μL of RPMI 1640 culture under gentle stirring by a rotational shaker. There were three conditions: cells in suspension without microspheres, cells with the moPLL/ALG microgel, and cells with moCHI/HEP microgel. The percentage of microspheres in relation to the total volume was 7%, and the suspension was agitated by a commercial rotational shaker (±60° oscillation at 25 rpm). A PicoGreen assay was performed on a Quant-iT™ PicoGreen^®^ dsDNA kit from Invitrogen to evaluate the culture’s cell proliferation.

### 3.7. Statistics

The results are provided as mean ± standard deviation (SD). The data were analysed, and outliers were identified using the ROUT method with a Q of 5%. After removing the outliers, the normality of the different samples was checked using the Shapiro–Wilk normality test with an alpha of 0.05. Unpaired T-Student tests (*p*-value = 0.05) were used to compare two single groups of data. An ordinary one-way ANOVA test (*p*-value = 0.05) was used for three or more groups to perform multiple comparisons between the column means when the normality test was passed. If this test was not passed, then the non-parametric Kruskal–Wallis test was used to compare this non-normal sample with other normal samples (*p*-value = 0.05) to perform multiple comparisons between the column means. GraphPad Prism 8 software (GraphPad Software, La Jolla, CA, USA) was used for the statistical analysis. Differences among groups were stated as *p* ≤ 0.05 (*), *p* ≤ 0.01 (**), *p* ≤ 0.001 (***), *p* ≤ 0.0001 (****) in normal and *p* ≤ 0.05 (#) in non-normal samples.

## 4. Results

### 4.1. Alginate Microsphere Production

The alginate microspheres (moALG) were produced in a flow-focusing microfluidic device according to the scheme in Figure 1. A 1% *w*/*v* aqueous solution of sodium alginate containing a dispersion of fine calcium carbonate particles was used as the discontinuous medium. The continuous medium in which microdroplets were formed was vegetable oil. In the second stage of the circuit, the outgoing oil flow with dispersed microdroplets of aqueous solution mixed with an external oil flow containing 2% *w*/*v* of acetic acid. Diffusion of acetic acid into the droplets causes calcium carbonate to release Ca^2+^ ions, which cross-link the alginate chains and form the gel microspheres, as has been described in Lui et al. [32]. It should be noted that in this procedure, the onset of ionic cross-linking occurs without the need for the microdroplets in the alginate solution to pass from the oily medium to an aqueous solution containing calcium ions, at which time the microdroplets tend to deform. As shown in the optical microscope images (Figure 3b), this form of “internal” cross-linking produces spherical microparticles. The effectiveness of the cross-linking process and consequently the formation of the alginate hydrogel in the microspheres are tested by transferring them to an aqueous medium in which they can retain their shape.

moALG’s mean diameter systematically decreases in an increasing discontinuous medium flow and depends to a lesser extent on the continuous medium (Figure 3a). Microspheres were obtained with diameters between 110–320 μm at discontinuous and continuous medium flows between 20–20 mL/h and 1–1.55 mL/h, respectively. An alginate flow of 2 mL/h yielded more significant microsphere diameter SDs and therefore was discarded. Table A1 in Appendix A gives the numerical data, including mean values and the SD referred to in Figure 3a. This behaviour agrees with other studies [33] since it is the force at which both fluids intersect to form the jet that regulates the size of the resulting microdroplets. For the rest of the work, the selected discontinuous and continuous flow values were 40 and 1.55 mL/h, respectively. The extra distance travelled by the moALG from leaving the microfluidic system until they reach the collection vessel helps to prevent them from aggregating during cross-linking.

CryoFESEM can show images of moALG cryo-fractured at −80 °C. The water contained in the sample crystallises on cooling and pushes on the alginate chains. After sublimating the water crystals, these trabeculated cores can be seen if the moALG is properly cut (Figure 3c). As the microspheres were formed from a 1% sodium alginate solution in water, a very small fraction of the volume was occupied by the alginate chains. The fine surface layer which can be observed in Figure 3d could also have been compacted by the crystallisation of the water the microsphere contained.

### 4.2. Polyelectrolyte Multilayer Coatings

The microspheres’ coating serves a double purpose in microgel cell cultures. Firstly, it should improve microsphere stability in the culture medium. This ability is described in Section 4.4, including the time dependence of the average microsphere diameter immersed in different aqueous media. Secondly, the surface must exhibit functional sequences capable of being recognised by cell receptors or capturing and presenting growth factors or cytokines to cells, as in the case of HEP, which plays an important role in sequestering growth factors from the medium and making them available to the cells. The presence of the biomolecules on the surface is described in Section 4.3.

Three bilayers of the chitosan/heparin system were deposited in one of the experimental series (here in after moCHI/HEP). At acidic pH, CHI acts as a polycation by protonating its amine groups and can be attached on the anionic surface of the moALG. After the CHI layer, the HEP layer was deposited, acting as a polyanion. This sequence was repeated three times, ending with a HEP layer [27]. Finally, the coating was cross-linked using the EDC/NHS chemistry [28]. The LbL procedure presents complications when applied to microspheres since they tend to agglomerate due to their surface electrostatic adhesion, which prevents high numbers of layers being achieved. This problem is aggravated in CHI by the difficulty of extracting the unbound chains from the surface in the intermediate washing steps because of the low solubility of CHI, even with water at pH 5.5.

Another series of samples was prepared with a coating of two bilayers of polylysine/alginate (moPLL/ALG). In this case PLL acts as polycation and alginate as polyanion [29].

Figure 4 shows cryoFESEM images of both types of coated microspheres in this work. The sections of the moCHI/HEP and moPLL/ALG microspheres are displayed in Figure 4a,b, respectively. While in the moPLL/ALG, an organised structure of the alginate trabeculae was observed, the moCHI/HEP microstructure was much more diffuse. The formation process of the LbL coating can explain the difference since the transfer of alginate microspheres from one aqueous medium to another implies a certain cross-linking density loss. In addition, the thin sub-micrometric layer of CHI/HEP can be seen in Figure 4d in an area of Figure 4c, where the coating has been lifted.

### 4.3. Coating Characteristics

The Blyscan test detects the presence of sulphated groups in the sample, thus allowing quantification of HEP in the moCHI/HEP coating. Although the moALG give a small signal attributed to the retention of reagents by the alginate gel, the difference in the coated samples (moCHI/HEP) is significant (Figure 5a). In fact, the moCHI/HEP amount to a concentration of 8.91 ± 2.23 μg HEP/mg when kept in water.

A ninhydrin assay was performed to quantify the PLL on the moPLL/ALG surface and determined a total of 126.30 ± 30.48 µg of PLL/mg of microspheres. In this case, the alginate gel (moALG) did not retain any reagents, as can be seen in Figure 5c.

Figure 5b shows the spectra of the pristine materials used to obtain the three types of microspheres studied in this work, while Figure 5d highlights the characteristic peaks in the FTIR spectrum of the wet microspheres themselves. Moist microspheres imply that the specific peaks of the materials will be attenuated in the spectra. Indeed, the microspheres obtained contain approximately 99% water, as previously mentioned. Nevertheless, certain things can be stated about these spectra: first, the characteristic peaks of the moALG are also detected in all the coatings. The peak in the pure alginate and the moALG at 1620 cm^−1^ is related to the symmetrical ketone bonds present in the ALG [34,35]. The shift of this peak to 1632 cm^−1^ in the coated microspheres is significant. In the case of moCHI/HEP, the displacement may be due to the overlapping peak of the C=N bond formed at the cross-linking points between the CHI’s amine group and the HEP’s carboxyl group [36]. In the case of moPLL/ALG, it is due to overlapping with the amide bond present in the PLL [37]. Besides, significant peaks were found between wavenumbers 1130–1000 cm^−1^. In moALG there are three very clear peaks at 1112, 1075 and 1018 cm^−1^ due to the vibrations of the carbonyl and ether groups [34,37]. In moPLL/ALG the peaks are also related to the same groups as in moALG but with a different distribution. Relevant variations are found in the moCHI/HEP spectrum, where peaks overlap to form a plateau at 1094–1072 cm^−1^. This spectrum area seems to indicate that the signal of the sulphate groups of HEP and the asymmetric ether stretching of ALG have overlapped [37].

moALG are able to absorb more than 3700% of water measured on a dry basis. In the thermogravimetry tests carried out under a nitrogen atmosphere, the sample lost most of its weight during water evaporation, which takes place at approximately 160 °C (Figure A1a in Appendix A). At 200 °C, the sample can be considered completely dry. The weight at 200 °C served to represent the thermograms based on the dry weight of the sample, as shown in Figure 6. The TGA thermogram of dry moALG presents two different degradation stages. One between 200–350 °C, where most weight loss occurs, and another at a higher temperature, from 350–1000 °C, where several smaller degradation steps can be observed, leaving a final residue of 18% (by weight). Similar thermal degradation behaviour is reported in [38,39].

Alginate microspheres lost 40–50% of their dry weight between 200–350 °C. Two peaks can be seen in the weight derivative *dw/dT* (Figure A1b) at 219 and 271 °C. A double peak during thermal degradation was reported on calcium alginate [39,40] and alginic acid [41]. Interestingly, in ALG metal salts only one peak tends to appear in this region or, at least, the two peaks get so close together that one of them appears as a shoulder on the high-temperature side of the main peak, as in the case of the thermograms of iron [38], copper [42] and sodium alginate [38,39]. The thermal degradation mechanism at this stage is associated with decarboxylation and fracture of glycosidic bonds [38].

In the moPLL/ALG microspheres, the two alginate peaks come together until they almost become a single one (Figure 6c,d). Pristine PLL’s main degradation peak in the weight derivative curve is at 332 °C (Figure 6c). To estimate the weight loss at this stage of moPLL/ALG thermal degradation, the *dw/dT* curve can be fitted to a Gaussian, as seen in Figure 6d. The area of the Gaussian determined that the fraction of weight loss associated with this individual contribution to degradation is 74% of the total weight. In the moPLL/ALG microsphere thermogram, the same peak is clearly observed in *dw/dT*. Proceeding in the same way, it is determined that the weight loss associated with this process is 11.5%, which provides an estimation of a PLL fraction of 15.5% (11.5/0.74%) in the coated microspheres. Since the alginate microspheres are made from a 1% alginate solution, the ALG mass in the coated microsphere is very small, so that the coating, despite being made up of molecular layers, represents an appreciable mass fraction. The change in the main shape of the alginate degradation with respect to the uncoated microspheres indicates that a strong interaction with the PLL should occur, especially in the LbL coating in which ALG layers are deposited between the PLL layers due to a strong electrostatic interaction. However, a certain PLL penetration into the microsphere’s core due to its high porosity cannot be ruled out. It must also be considered that the alginate’s main degradation occurs at temperatures below the PLL degradation temperature, which means that the volatile substances released during ALG degradation must go through the coating to get out of the material and that their loss is detected by the thermobalance. The diffusion of these substances could also significantly affect the thermogram shape.

In moCHI/HEP microspheres, where ALG is present only in the microsphere core, a shift can also be seen in the degradation peak towards high temperatures. Gaserød et al. showed the easiness with which the CHI coating diffuses into the ALG core, depending on several factors, such as alginate porosity, molecular weight and others [43]. The interaction between CHI and ALG could be responsible for this behaviour. However, as mentioned above, this effect is smaller than when part of the ALG alternates in the coating with PLL layers, in which case it is not possible to quantify the CHI or HEP fraction in the LbL-coated microsphere with TGA because the degradation peaks of the three components are very close to each other (Figure 6b). Finally, an ash fraction of the order of 25% remains above 1000 °C.

The ALG chain degradation in the temperature interval between 200–350 °C leads to intermediate, more stable and slower degradation products [38]. In this region, between 350–1000 °C, pure alginate microspheres show two broad peaks in *dw/dT* around 500 °C and 800 °C. These two processes are also detected in the coated microspheres, but shifted in temperature and intensity, as expected, due to the complete change in the material’s microstructure after the first stages of degradation [39]. HEP presents a sharp peak at 730 °C in this zone, which is not noticeable in the coated microspheres, although a broader peak is seen around 670 °C that does not appear in moALG (Figure 6a,b). Finally, an ash fraction of the order of 25% remains at 1000 °C

### 4.4. Equilibrium Swelling

The alginate hydrogel formed by ionic cross-links tends to progressively lose cross-link density on immersing the gel in an aqueous medium. The tendency of calcium ions to migrate from the hydrogel into the medium is governed by thermodynamic equilibrium criteria. When the hydrogel cross-linking density decreases, its swelling capacity increases [44], which in our microspheres means a larger diameter, while polymer chains may be delivered and dissolve in the medium so that finally the whole microsphere may disappear. In this study, the microsphere stability was determined by measuring the average diameter after immersion in different commonly used aqueous solutions. moALG microsphere water absorption capacity has been estimated at 37.7 ± 2.5 g(water)/g(ALG) on a dry basis through gravimetric tests. After immersion in different liquids, an optical microscopy analysis was performed to compare the effect of the different aqueous media in which they had been immersed. The moALG, moCHI/HEP and moPLL/ALG samples were initially kept equilibrated in ultrapure water at pH 7 after which they were transferred to different media, when images were taken at different times.

Figure 7a shows the values of moALG mean diameter measured 30 min after immersion in the liquid media. It can be seen that changes in pH do not significantly affect the size of the microsphere, but immersion in saline culture media increases their water absorption capacity by up to 60%. An EDTA solution was also used as a control which, as expected, completely dissolved the moALG microspheres in the first few minutes (data not shown).

moALG change in size over time in certain media. Again, variations in microsphere diameter occur only in saline culture media (Figure 7b). In some cases, average particle size increases after immersion, e.g., DMEM R. This is attributed to the progressive cross-linking density loss due to calcium ion migration into the medium. In other culture mediums, such as RPMI 1640, PBS+/+, and PBS−/−, the particle size decreases, so that alginate chains are already being released. After 14 days of immersion, the diameter can shrink by up to 40% (Figure 7b). Table A2 in Appendix A collects the numerical data, including the mean values and SD included in Figure 7a,b.

Surface coatings aim to stabilise microsphere dimensions over time during cell cultures. Both moCHI/HEP and moPLL/ALG coatings effectively stabilised the microspheres in EDTA by maintaining their integrity over time (Figure 8a,c). Although microcapsules formed from liquefied alginate mostly remain intact for up to 14 days, some tend to deform, leading to wrinkling (see Figure 8a).

The swelling capacity of microspheres coated with polyelectrolytes changes significantly with respect to moALG. At time 0, diameters were larger in moCHI/HEP than in moALG when immersed in culture media (Figure 8b), leading us to believe that calcium ions were being released during the LbL deposition process, reducing cross-linking density and increasing their water absorption capacity. In contrast, the difference between the diameter of moPLL/ALG microspheres and that of moALG is much smaller than in the case of moCHI/HEP in most conditions (Figure 8b). Interestingly, there were no significant differences in the moCHI/ALG microsphere diameters, either with changes in the liquid medium or time (Table A3 in Appendix A). In moPLL/ALG, there were minor differences between the microspheres at pH 7 and other conditions at time 0, although, over time, the sizes remained unchanged in each condition (Table A4 in Appendix A). The results obtained are a consequence of the thermodynamic equilibrium criteria, which equalise the chemical potential of the water inside the microsphere and in the liquid medium. The activity coefficient of water, and therefore its chemical potential, is different in the various saline media used. The chemical potential of water in the moALG depends on both the entropic forces that tend to expand the network (and in the absence of cross-linking points would tend to dissolve the polymer) [45] and the elastic forces that tend to oppose swelling [44]. The coating produced by LbL forms a membrane around the surface of the microsphere, as shown by introducing the microspheres in EDTA, while if the coating is not cohesive the microsphere dissolves, as occurs with uncoated moALG. The results in Figure 8b,d and Table A2, Table A3 and Table A4 in Appendix A show that both coatings generate sufficient elastic energy so that changing the microsphere from the medium at pH 7 to a saline medium such as PBS−/− does not significantly increase the diameter, as in the moALG. Another important aspect of using these microspheres to form microgels is that they remain cohesive over time, even with the gradual reduction of the cross-linking density of the alginate network that forms their body (see Figure 7).

### 4.5. Cell Culture in the Microgel

Using in vitro cultures to study tumour cell behaviour requires reproducing the tumour microenvironment as accurately as possible. Blood cancer cell sensitivity to antitumor drugs may be completely different in monolayer or suspension cultures than in three-dimensional environments [46]. The response to a drug can also greatly depend on the presence in the environment of the cells of other cellular or non-cellular components present in the biological niche. In this regard, the coatings described in this paper allow cells to be provided with the biomolecules of interest to either recreate a biological environment or present cell adhesion sequences. Using the RPMI 8226 cell line, we compared the evolution of the total DNA content in both suspension cultures and in cultures with the two microgels under study (moCHI/HEP and moPLL/ALG). The total DNA data were then used to calculate the approximate number of cells and the cell proliferation rate by means of the appropriate calibration procedure [47].

The RPMI 8226 cells and the microspheres (moCHI/HEP and moPLL/ALG) were suspended in RPMI 1640, the default culture medium for multiple myeloma cell culture. Suspension homogeneity was maintained by gentle agitation so that the cells always had functional surfaces at a distance of the order of their cell sizes. The numbers of cells were quantified as a function of time, based on the measurement of total DNA and the suspension culture without microspheres, and the same agitation system was used as a control. We tested the cultures for up to 5 days, as for longer times the number of cells per millilitre of culture medium would have exceeded the supplier’s recommendation for this cell line. The result shown in Figure 9 confirms the proliferation of cell cultures in the microgel and suspension. The absence of statistically significant differences between the control and the microgel samples indicates that our platform does not change the cell behaviour and paves the way for further studies on the role of the different components of the bone marrow extracellular matrix in the progression of the disease.

## 5. Conclusions

Alginate hydrogel microspheres immersed in the usual cell culture media gradually lose cross-linking density and release part of their polysaccharide chains into the medium. The polyelectrolytes multilayer coating of CHI/HEP or PLL/ALG thus formed stabilises the microsphere dimensions in all the culture media tested, as can be seen by the evolution of the mean particle diameter when immersed in an aqueous medium. The LbL process makes it possible to form a cohesive coating that survives even when the gel is liquified inside it. However, part of the ionic cross-links can be lost during the formation of the LbL multilayer, which accounts for the greater average diameter of moCHI/HEP than that of moPLL/ALG when immersed in water at pH 7. On the other hand, the elastic forces exerted by the multilayer coating hinder gel swelling, and the average diameter remains unchanged with immersion time, even in the saline media in which the uncoated alginate swells significantly more than in pure water. The microgel formed by the suspension of microspheres in the liquid medium generates a three-dimensional environment suitable for multiple myeloma cell cultures while maintaining its capacity to proliferate. This system can thus be considered as a promising semi-solid culture medium for tumoral blood cancer cells.

## Figures and Tables

**Figure 1 polymers-14-04282-f001:**
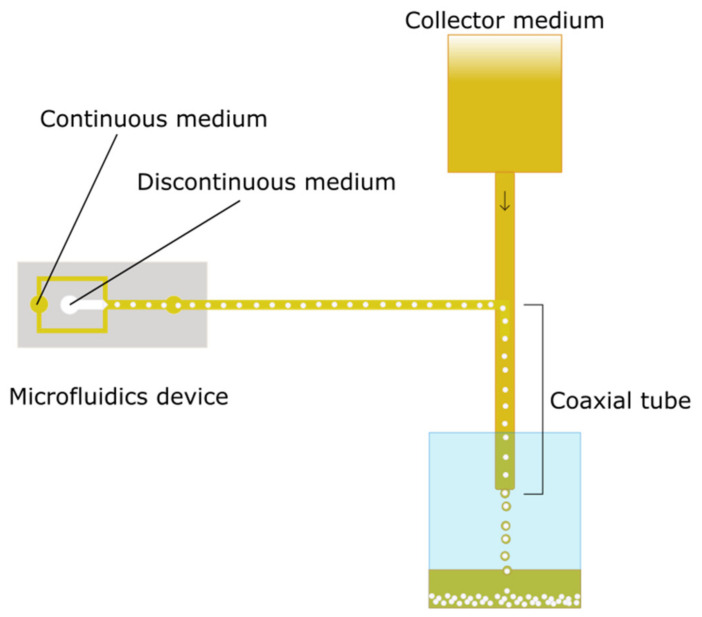
Diagram of microfluidics-based system used to obtain alginate microparticles.

**Figure 2 polymers-14-04282-f002:**
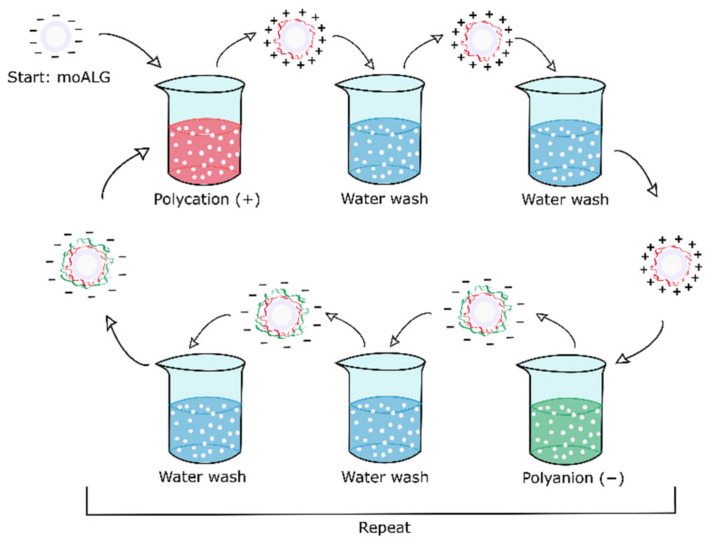
Scheme of the immersion LbL technique used to functionalise the microspheres. The process is repeated until the desired number of layers is achieved.

**Figure 3 polymers-14-04282-f003:**
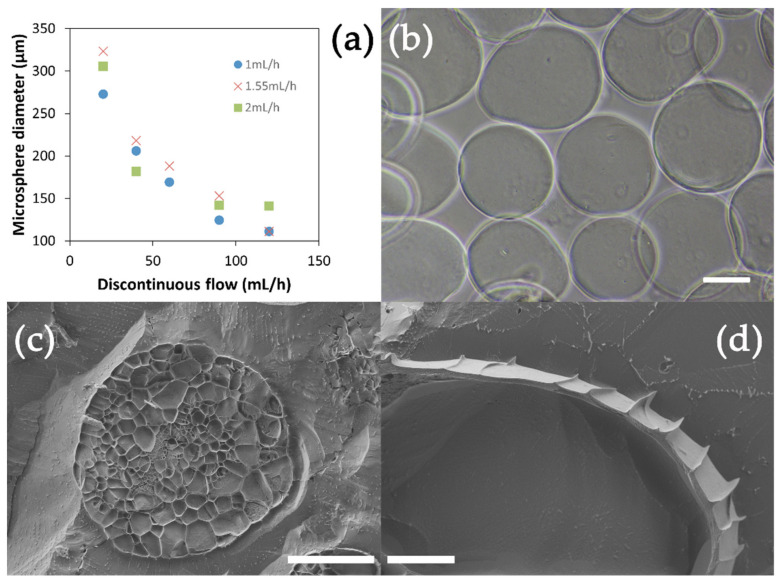
(**a**) Alginate microspheres (moALG) mean diameter as a function of the flow rate of the continuous medium. (**b**) Appearance of moALG in optical microscopy (scale bar 100 μm). (**c**) Cryo-FESEM images of a cryo-fractured moALG (scale bar 100 μm) and (**d**) detail of the external surface of a cryo-fractured moALG (scale bar 10 μm).

**Figure 4 polymers-14-04282-f004:**
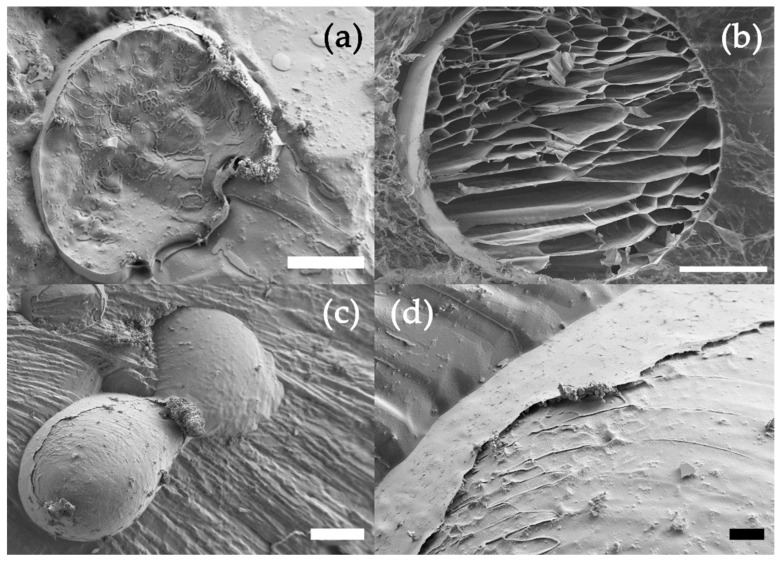
(**a**) CryoFESEM images of cryo-fractured moCHI/HEP and (**b**) moPLL/ALG microspheres (scale bar 50 μm). (**c**) shows two adhering microspheres moCHI/HEP due to the LbL coating (scale bar 100 μm), while (**d**) shows a detail of the coating in moCHI/HEP (scale bar: 10 μm).

**Figure 5 polymers-14-04282-f005:**
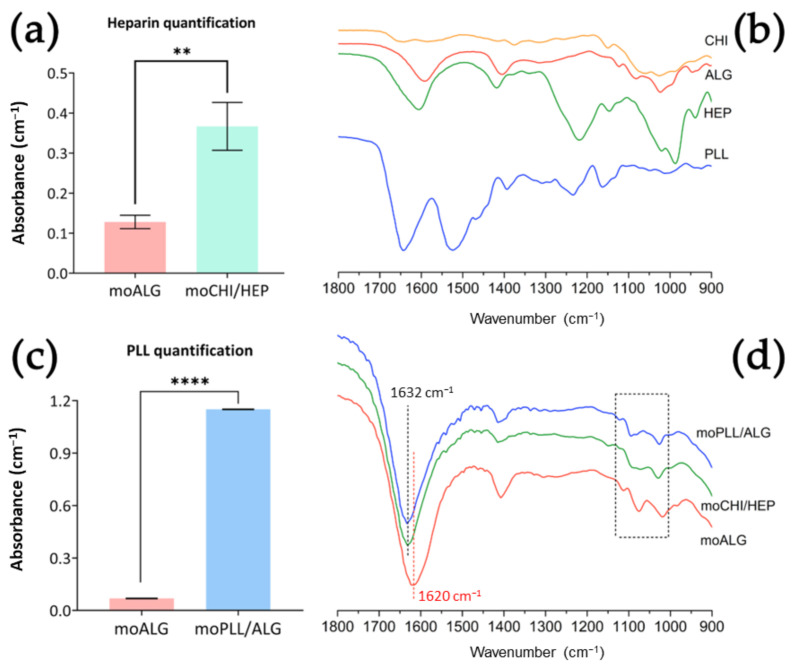
(**a**) Quantification of HEP in moCHI/HEP. (**b**) FTIR spectra of the pristine components of the several types of microspheres obtained. (**c**) Quantification of PLL in moPLL/ALG. (**d**) FTIR spectra of the different types of microspheres obtained. Comparisons between different types of microspheres were made using an unpaired T-Student test. *p*-value legend: *p* ≤ 0.01 (**), *p* ≤ 0.0001 (****).

**Figure 6 polymers-14-04282-f006:**
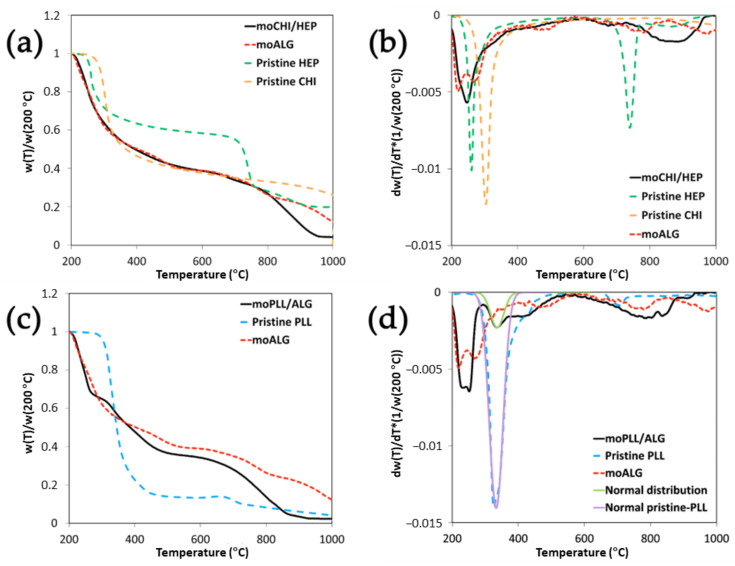
TGA thermograms of LbL-coated alginate microspheres with different polyelectrolyte pairs. The values of weight and derivative of the weight with respect to temperature have been normalised with respect to the value of the weight of the sample at 200 °C when the sample is completely dry. (**a**,**b**) represent the weight and derivative of the weight of the moCHI/HEP and the moALG with respect to temperature. (**c**,**d**) represent the same data for moPLL/ALG. The thermograms of the pure components are shown for reference.

**Figure 7 polymers-14-04282-f007:**
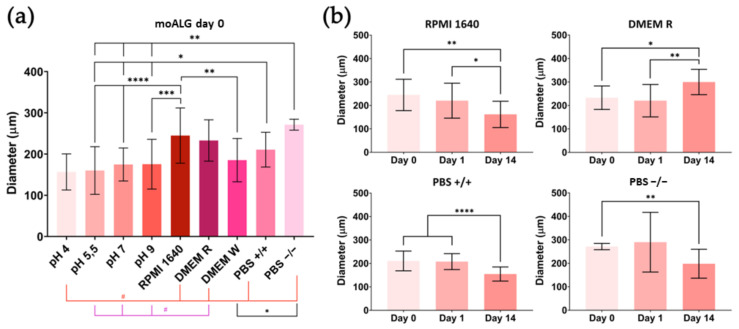
Numerical data of equilibrium swelling experiment on moALG: (**a**) diameter of moALG microspheres immersed in different aqueous media at day 0; (**b**) time dependence of microsphere size. Mean ± Standard Deviation. Comparisons between different conditions were made using an ordinary one-way ANOVA test or a non-parametric Kruskal–Wallis test, according to the normality of the data obtained. Differences were stated with the *p*-value: *p* ≤ 0.05 (*), *p* ≤ 0.01 (**), *p* ≤ 0.001 (***), *p* ≤ 0.0001 (****) in normal and *p* ≤ 0.05 (#) in non-normal samples.

**Figure 8 polymers-14-04282-f008:**
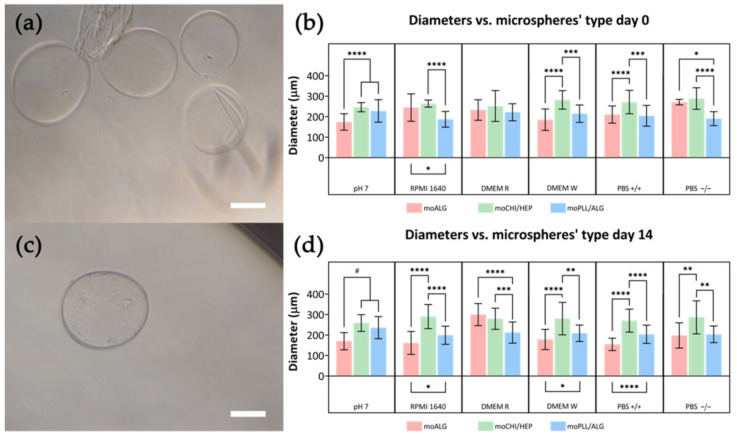
Equilibrium swelling of moCHI/HEP and moPLL/ALG microspheres in different media at 0 (**b**) and 14 days (**d**). Images of the microspheres immersed in an EDTA solution at 14 days are shown: moPLL/ALG (**a**) and moCHI/HEP (**c**) (scale bar 100 µm). Comparisons between different conditions were made using an ordinary one-way ANOVA test or a non-parametric Kruskal–Wallis test, according to the normality of the data obtained. Differences were stated with the *p*-value: *p* ≤ 0.05 (*), *p* ≤ 0.01 (**), *p* ≤ 0.001 (***), *p* ≤ 0.0001 (****) in normal and *p* ≤ 0.05 (#) in non-normal samples.

**Figure 9 polymers-14-04282-f009:**
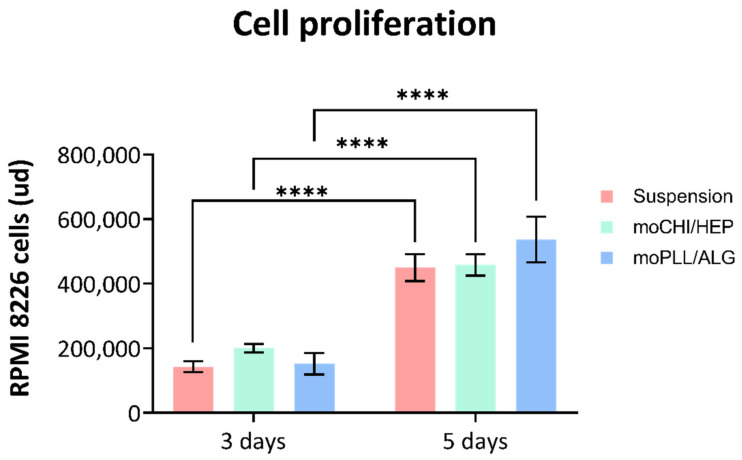
Proliferation of RPMI 8226 multiple myeloma cells in the microgels at 3 and 5 days (**** *p* < 0.0001).

## Data Availability

Publicly available datasets were analyzed in this study. This data can be found at Riunet repository of Universitat Politècnica de València, here: [http://hdl.handle.net/10251/187455].

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
