# Peer review of "Stability of Biomimetically Functionalised Alginate Microspheres as 3D Support in Cell Cultures"

_polymers, 2022, doi:10.3390/polym14204282_

Round 1

Reviewer 1 Report (Previous Reviewer 1)

The authors responded point by point to the reviewer's comments by covering the shortcomings of the manuscript. Therefore the manuscript should be considered for publication.

Author Response

Thank you very much for your comments

Reviewer 2 Report (Previous Reviewer 2)

I am still not satisfied with the English language

Author Response

Thank you very much for your comments. We have had the text checked again by a specialist in proof-reading scientific texts, the changes made in the text are marked with the Word track changes.

Reviewer 3 Report (Previous Reviewer 3)

The revised manuscript has been improved and can be accepted for publication.

Author Response

Thank you very much for your comments. The number of references has been reduced. 

This manuscript is a resubmission of an earlier submission. The following is a list of the peer review reports and author responses from that submission.

Round 1

Reviewer 1 Report

The submitted manuscript entitled " Stability of biomimetically functionalised alginate microspheres as 3D support in cell cultures " by García-Briega et al., shows  alginate microspheres are functionalised by coating their surface with  biomolecules from the extracellular matrix of the tissue of interest to be used as a 3D support for cell culture. The presented work is well written and shows stability of biomimetically functionalised alginate microspheres. however, there are some issues to be addressed.

Below is some points where authors need to modify their manuscript for more good presentation for the readers.  

1. If possible, the author should show the Layer by Layer functionalisation in a diagram, which is more conducive to the reader's understanding of the preparation process.

2. The characteristic peaks of the FT-IR spectrum of Figure 4 are not very obvious, especially the PLL of Figure 4B, it is recommended to re-test.

3. Authors need to add more discussion for Fig.7. Not only results. 

4. Conclusions is not satisfactory and it must need to elaborate more with future perspectives.

5. Correct the References using the guide of Journal.

Reviewer 2 Report

The study entitled “Stability of biomimetically functionalised alginate micro-
spheres as 3D support in cell cultures is based on the evaluation of the properties of alginate microspheres as 3D support in cell cultures. The overall impression of this manuscript is good. The results have been thoroughly discussed with good justification. The references provided fully support the given text. However, there are still some problems which must be corrected before the acceptance of this manuscript. I therefore, suggest minor revision at this stage.

1) The authors are being encouraged to answer what was their motivation in choosing and arriving at the idea of using alginate microsphere for this purpose as there have been many other methods tried so far to cope with this problem.  

2) The authors are being encouraged to put a paragraph discussing the challenges being faced upon using alginate microsphere in the cell culture. For instance, issues related to their availability, cost, limitations to specific natural environment, compatibility, sustainability and so on.     

3) The authors are encouraged to add the following most relevant reference in the reference section.

i) M. Khan, L. Tiehu, S.B.A. Zaidi, E. Javed, A. Hussain, A. Hayat, A. Zada, D. Alei, A. Ullah, Synergistic effect of nanodiamond and titanium oxide nanoparticles on the mechanical, thermal and electrical properties of pitch derived carbon foam composites,            Polym. Int. 70 (2021) 1733-1740.

ii) A. Hamid, M. Khan, F. Hussain, A. Zada, T. Li, D. Alei, A. Ali, Synthesis and physiochemical performances of PVC-sodium polyacrylate and PVC-sodium polyacrylate-graphite composite polymer membrane, Z. Phys. Chem. 235:12 (2021) 1791-1810.

iii) M. Khan, L. Tiehu, A. Hussain, A. Raza, A. Zada, A. Dang, A. R. Khan, R. Ali, H. Hussain, J. Hussain, Z. Wahab, M. Imran, Physiochemical evaluations, mechanical attenuations and thermal stability of graphene nanosheets and functionalized nanodiamonds loaded pitch derived carbon foam composites, Diam. Relat. Mater. 126 (2022) 109077.

iv) J. Raza, A. Hamid, M. Khan, F. Hussain, A. Zada, L. Tiehu, A. Ali, P. Fazil, Z. Wahab, Preparation and comparative evaluation of PVC/PbO and PVC/PbO/graphite based conductive nanocomposites, Z. Phys. Chem. (2022), https://doi.org/10.1515/zpch-2022-0051

4) It has been thoroughly observed that the tables and figures have not been properly referred in the main text.  

5) The text given in Line-99, Page-3 stating, “Different coatings will be used to functionalise the microspheres” needs to be replaced by the correct line stating, “Different coatings have been used to functionalise the microspheres”.

6) English language of the manuscript needs to be improved.

Reviewer 3 Report

The submission (ID: polymers-1929613) explored the stability of alginate microspheres functionalized by grafting specific biomolecules onto the surface of alginate microspheres. However, the content and innovation of this manuscript cannot meet the requirements of the journal. In a word, the quality of this article can not meet the requirements of this journal, and there are some issues that need to be clarified before submitting the manuscript to other journals.

1.     The references in the introduction are too old and lack of up-to-date literature, especially in the last five years.

2.     The logic of the manuscript is not strong, and there are format problems, such as lines 73-74, 80-81, 349-350, 358-359, and all the symbols of “degrees Celsius”.

3.     Lines 203-207: please provide evidence of the proof!

4.     Lines 237-238: please provide evidence that improves the stability of the microspheres in the culture medium!

5.     Lines 252-254: "However, as seen below……the heparin on its surface", no data and evidence of "all are sufficient" have been found. Please explain?

6.     Line 264-265: since the manuscript considers it meaningful to study the stability of alginate microspheres in different aqueous solutions, please provide the results?

7.     Lines 424-427: data on the activity and proliferation of tumor plasma cells are not shown.

8.     Figure 8: it is recommended to provide more data on the number of culture days to fully analyze the viability of cells.

9.     The manuscript focuses on the stability of alginate microspheres, but the stability data are not enough in the manuscript data.

10. Line 450-451: no evidence of "multiple Myeloma cells are viable" was found in the manuscript, please point out or provide!